# Development of a validated PROM set for children with tinnitus using the Patient-Reported Outcomes Measurement Information System – Protocol for the 'Tinnitus in Children' (TinC) study

**Denise Fuchten**[1,2], **Inge Stegeman**[1,2], **Adriana L. Smit**[1,2]*

**1** Department of Otorhinolaryngology, Head and Neck Surgery, University Medical Center Utrecht, Utrecht, The Netherlands, **2** UMC Utrecht Brain Center, University Medical Center Utrecht, Utrecht, The Netherlands

* a.l.smit-9@umcutrecht.nl

## Abstract

### Introduction

Tinnitus in children remains an underexplored area of research, with limited knowledge about its prevalence and impact on daily lives. Current assessments of the impact of tinnitus in children predominantly rely on open-ended questioning during clinical interviews, inconsistent use of structured tools, and data reported by adult proxies. These methods often fail to fully capture the child's personal experience of tinnitus, underscoring the need for a standardized, child-centered measure to more accurately assess its impact and improve understanding of children's experience. To address this gap, this study proposes the development of a validated PROM set using the Patient-Reported Outcomes Measurement Information System (PROMIS).

### Methods

The proposed study is a monocenter, mixed-methods, multi-phase observational study. The study will consist of six consecutive steps: (1) identifying patient-important outcomes through focus groups with children aged 8–18 years; (2) defining the underlying constructs of the outcomes and linking them to relevant PROMIS domains; (3) gathering input form an expert panel to refine the outcomes and domains; (4) discussing the PROM set with the focus groups and assess its feasibility; (5) assessing content validity in the expert panel; and (6) assessing the reliability in a sample of children with tinnitus.

**Data availability statement:** No datasets were generated or analysed during the current study. All relevant data from this study will be made available upon study completion.

**Funding:** This project is partially funded by the Dorhout Mees Stichting and through crowdfunding by Stichting Oorfonds (UMCU budgeted grant funding number 76231.92). However, the funders did not play any role in the study design and preparation of the manuscript. There was no additional external funding received for this study.

**Competing interests:** The authors have declared that no competing interests exist.

## Discussion

By using the generic PROMIS framework to develop a condition-specific measure, the proposed study aims to create a standardized, child-centered PROM set that captures the specific impact of pediatric tinnitus, while maintaining the ability to compare outcomes across different conditions and populations. This PROM set could ultimately be used in both clinical care and research to evaluate tinnitus impact and assess the effects of treatments.

## Introduction

Over the years, research on tinnitus has predominantly focused on adults, leaving studies on the pediatric population relatively scarce [1]. While estimates of tinnitus prevalence in children vary widely, studies indicate it could affect between 4.7% and 46% of those with normal hearing and between 23.5% and 62.2% of those with hearing loss, depending on the population, study design, and definition used [1]. There is still a significant gap in our understanding of tinnitus and its impact on children, but existing studies suggest that the sound children experience can be responsible for difficulties in concentrating as well as in hearing and listening, which in turn can cause problems at school and in other activities. Furthermore, tinnitus can impact their sleep, lead to a worsening of their mental health, and result in social isolation [2]. However, the assessment of the experienced tinnitus-related problems in these studies too date largely relied on open-ended questioning during clinical interviews and inconsistent use of structured questions, with much of the data reported by adult proxies [2]. Additionally, it is important to note that children are less likely to spontaneously report tinnitus and associated problems [3,4], and when they do, they often struggle with describing these abstract concepts [5]. This underscores the need for a more standardized, child-centered approach to measuring the impact of pediatric tinnitus in both clinical care and research.

Patient reported outcome measures (PROMs) have become an essential tool for understanding and quantifying the subjective experiences of individuals with varying health conditions. Generic PROMs, such as the Pediatric Quality of Life Inventory (PEDsQL) [6], provide a broad assessment of a child's overall well-being and allow for comparisons across different conditions and populations. However, their broad nature may mean they lack sensitivity to measure the specific symptoms and challenges associated with particular conditions, such as tinnitus. While condition-specific PROMs lack the generalizability of generic PROMs, they are more sensitive to capture these experiences relevant to a particular condition or population [7]. For adults with tinnitus, multiple of these condition-specific PROMs are available, such as the Tinnitus Functional Index (TFI) [8] and the Tinnitus Handicap Inventory (THI) [9]. While these PROMs are sometimes used to assess tinnitus in children [2,10], they were not developed with the intention to be used in this population and have not been validated for this purpose [10]. Therefore, a condition-specific PROM for children with tinnitus, the Impact of Tinnitus in Children Questionnaire (iTiCQ) [11], has recently

been developed. This self-report tool for children aged 8–16 years aims to address the impact of tinnitus in children, though further validation and translations are necessary to establish its effectiveness and accessibility across different populations.

A more comprehensive and innovative method for assessing health outcomes involves integrating generic and condition-specific approaches in PROMs. This method enables the development of tools that can accurately assess condition-specific impacts, while maintaining the ability to compare outcomes between different conditions and populations. This integrated approach is especially advantageous for conditions with multiple comorbidities, such as tinnitus [12,13], as the experiences of patients do not have to be attributed to a single health condition [14].

The Patient-Reported Outcomes Measurement Information System (PROMIS) offers a framework to enable this integration of generic and condition-specific approaches in health assessment. The PROMIS is a dynamic measurement system comprising generic item banks developed to assess physical, mental and social health outcomes in both adults and children [15,16]. An item bank is a collection of items that measure the same construct, or domain, such as sleep disturbance, depression, or peer relationships. These items are developed based on existing PROM items, which are evaluated, standardized and methodologically tested before they are included in the PROMIS item bank [17–19]. The constructs measured by these item banks can be assessed in a variety of languages using short forms or Computerized Adaptive Testing (CAT), with scores provided on a T-score metric. This allows for comparison with the general population, as well as between populations and conditions [16]. The dynamic character of the PROMIS enables flexibility in selecting item banks relevant to specific research goals or clinical applications, thereby facilitating a tailored approach to health assessment and providing the ability to use a generic framework to facilitate condition-specific assessment.

The objective of the proposed study is therefore to develop a PROM set for children with tinnitus using the PROMIS methodology. By combining input from patients with expert opinions, the study aims to create a list of outcomes that accurately reflect the impact of tinnitus on children of different ages and select relevant domains from the PROMIS framework to develop a validated PROM set. This approach ensures that the measure captures the specific impact of pediatric tinnitus while utilizing the generic framework of PROMIS for broader applicability and comparability.

## Methods

### Study design

The study is a monocenter, mixed-methods, multi-phase observational study conducted at the Wilhelmina Children's Hospital (Wilhelmina Kinderziekenhuis; WKZ), part of the University Medical Center (UMC) Utrecht, a tertiary referral clinic in the Netherlands.

The study is organized into 6 consecutive steps (Fig 1), with the objective to develop a PROM set for children with tinnitus using the PROMIS framework. The study starts with identifying patient-important outcomes for pediatric tinnitus trough focus groups with children. The underlying constructs of these outcomes will then be defined and linked to relevant PROMIS domains. The defined outcomes and associated domains will be discussed with an expert panel as well as with the focus groups, after which necessary adjustments will be made to the measure. The last steps will consist of assessing the psychometric qualities of the PROM set by assessing the content validity in the expert panel and the reliability in a sample of children with tinnitus. The methodology for each step of the study is detailed in the following sections.

### Step 1. Identify patient-important outcomes for children with tinnitus in focus groups

**Participants.** Children with tinnitus, ranging from 8–18 years old, will be included in this step of the study. Participants will be divided in focus groups based on their age; 8–11, 12–15 and 16–18 years old. Each focus group will consist of six children to ensure a diverse range of perspectives while maintaining a structured environment where participants feel comfortable sharing their thoughts and experiences [20]. As many focus groups per age category will be held until data

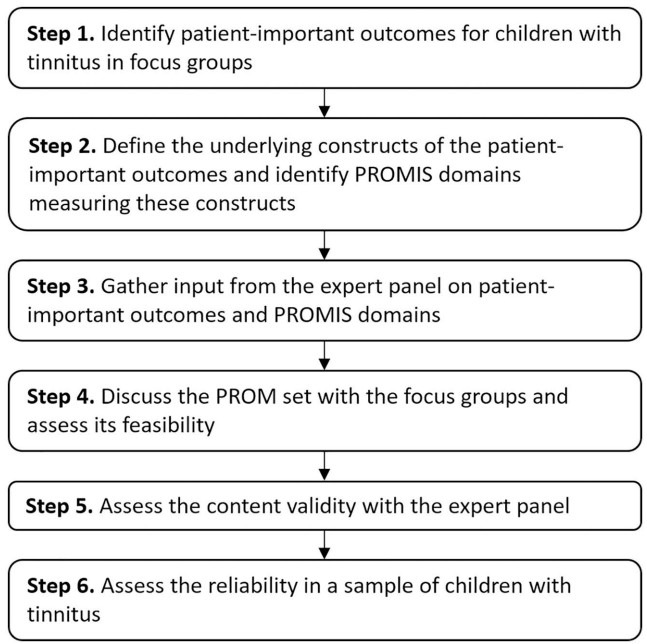

**Fig 1. Study design.**

saturation is reached for listed patient-important outcomes. For this, a minimum of two focus groups per age category are required, and consequently, a minimum of 36 participants will be included during this step.

In order to be eligible to participate in the focus groups, a potential participant must meet all of the following criteria:

- Between 8 and 18 years old at time of consent

- Presence of tinnitus for at least 3 months

- Willingness and ability to participate in focus group discussions

- Dutch language proficiency

- No diagnosed cognitive or developmental delays incompatible with age-appropriate functioning

The participants will be recruited at the otorhinolaryngology department of the WKZ and using flyers and online advertisements. If a participant is willing to participate and eligible based on the defined in- and exclusion criteria, the written patient information letter and informed consent will be sent. Dependent on the age of the participant, the following informed consent should be signed:

- 8–11 years old: Informed consent signed by a parent

- 12–15 years old: Informed consent signed by both the participant and a parent

- 16–18 years old: Informed consent signed by the participant

**Focus groups.** Each focus group will be conducted by a lead moderator and a co-moderator who are familiar with tinnitus care for children, and will take approximately 45 minutes. The lead moderator will guide the discussion and introduce topics when necessary. The co-moderator will take notes during the discussion and will step in with questions or clarifications to assist the moderator in interpreting the children's responses.

After a short introduction to the session, the discussion will start by asking the question 'Do you notice that, because of tinnitus, things go differently than they do in peers?'. After the introductory question, the discussion will be as flexible and open-ended as possible, so that all topics important to the participants will be covered and they will be steered as little as possible in their answers. When no more new topics are introduced by the children, the following topics will be introduced if not yet discussed: school/ability to concentrate, hearing ability and participating in conversations, sleep/fatigue, mental health, participating in (social) activities/relaxation, and relationships with friends and family. This list of topics is based on the concepts included in the PEDsQL, developed to measure health-related quality of life in children, the TFI, a distress questionnaire assessing tinnitus in adults, and the iTiCQ, a self-report measure of tinnitus impact for children.

**Analysis.** All focus group sessions will be audio recorded and transcribed verbatim. Two independent analysts will analyze the content of the transcriptions in an inductive manner, using the six stages of thematic analysis as described by Braun and Clarke (2006) [21], to identify patient-important outcomes per age group. NVivo analysis software will be used to facilitate the coding and organization of the data. Disagreements about the outcomes will be discussed in the research team to reach consensus. A final list of patient-important outcomes will be reported.

## Step 2. Define the underlying constructs of the patient-important outcomes and identify PROMIS domains measuring these constructs

The patient-important outcomes identified in step 1 will be reviewed to define their underlying constructs. These constructs will then be linked to the relevant domains with corresponding item banks in the PROMIS system. For this, two independent researchers will systematically assess the PROMIS database. Any disagreements regarding the domains will be discussed within the research team to reach consensus.

## Step 3. Gather input from the expert panel on patient-important outcomes and PROMIS domains

The patient-important outcomes, underlying constructs and corresponding PROMIS domains will be reviewed and refined based on the input from an expert panel, comprising key stakeholders involved in pediatric tinnitus care. This panel will include ENT surgeons, audiologists, psychologists, and adult patient delegates. As this panel will also be involved in the assessment of content validity in a later step, a minimum of seven members will be included, in line with the COnsensus-based Standards for the selection of health Measurement INstruments (COSMIN) guideline for evaluating the content validity of PROMs [22].

Experts will receive all relevant study information, including the patient-important outcomes identified in step 1 and the underlying constructs and corresponding PROMIS domains identified in step 2. They will be asked to provide individual feedback on whether any outcomes require refinement, merging, clarification, or if any important outcomes are missing. Based on the collective input, the research team will analyze the feedback and make the necessary adjustments to ensure that the PROM set incorporates the perspectives of all relevant disciplines.

## Step 4. Discuss the PROM set with the focus groups and assess its feasibility

In this step, the PROM set will be discussed with the same focus groups comprising children with tinnitus as described in step 1. During this discussion the aim is to gather feedback on the measure and to assess the feasibility, considering factors such as ease of use and time required for completion. The input from these focus groups will be used to further refine the PROM set, ensuring it is both practical and effective for assessing the impact of tinnitus in children.

## Step 5. Assess the content validity with the expert panel

The content validity of the PROM set will be assessed by the expert panel following the COSMIN guideline for evaluating the content validity of PROMs [22]. This assessment will focus on two key aspects: the relevance of the items and the

comprehensiveness of the PROM set. Since our target group consists of children, who may find it difficult to evaluate these aspects, the evaluation will be conducted exclusively within the expert panel. Comprehensibility, which is typically assessed as part of the COSMIN guideline for evaluating the content validity of PROMs, will not be evaluated in this study, as it has already been tested within the broader PROMIS system [23,24].

**Step 6. Assess the reliability in a sample of children with tinnitus**

In the final step of this study, the developed PROM set will be tested on reliability in a sample of 100 children with tinnitus, following the COSMIN recommendations [25]. Test-retest reliability will be assessed by having the children complete the PROM set twice, with a two-week interval. The intraclass correlation coefficient (ICC) will be used to evaluate the consistency of the measure over time. Additionally, the Standard Error of Measurement (SEM) and the Smallest Detectable Change (SDC) will be calculated to estimate measurement error and to identify the smallest change in score that reflects a true difference, rather than variability due to measurement error.

In order to be eligible to participate in this part of the study, a potential participant must meet all of the following criteria:

- Between 8 and 18 years old at time of consent

- Presence of tinnitus for at least 3 months

- Dutch language proficiency

- No diagnosed cognitive or developmental delays incompatible with age-appropriate functioning

Participant recruitment, eligibility assessment, and the informed consent procedure will be conducted in accordance with the procedures outlined in step 1. This includes recruitment at the Otorhinolaryngology department of the WKZ and via flyers and online advertisements, as well as an age-dependent informed consent procedure.

**Ethics and dissemination**

This study does not fall under the scope of the Dutch Medical Research Involving Human Subjects Act (WMO) as confirmed by the accredited medical ethics committee NedMec (protocol number 24–191/DB). It therefore does not require ethical approval from an accredited medical ethics committee in the Netherlands. However, an independent quality check has been carried out to ensure compliance with legislation and regulations (regarding Informed Consent procedure, data management, privacy aspects and legal aspects) in the UMC Utrecht.

The handling of personal data will comply with the EU General Data Protection Regulation and the Dutch Act on Implementation of the General Data Protection Regulation ('Uitvoeringswet AVG'). All data will be handled confidentially. All subjects will receive an unique identification code after informed consent. Names mentioned during the interviews will be pseudonymized in the transcripts using the identification code. Due to the personal and privacy sensitive nature of the (transcripts of the) interviews, sharing this data with external institutions will not be permitted. Data will be stored in a secure research folder structure with access control, which ensures only authorized members of the research team have access to personal data, including the key table that links personal data to the identification code. The expert panel will not have access to the research folder structure and/or files with personal data. Informed consent forms and paper files will be stored safely in a locked cabinet in a locked room in the study center.

The study results will be made accessible to the public in a peer-reviewed open access journals.

**Trial status**

This study is currently in the recruitment phase. Participant recruitment, data collection, and analysis will take place in multiple phases and are expected to be completed by the end of 2027, depending on the pace of recruitment and logistical factors.

## Discussion

To address the need for more standardized, child-centered instruments for assessing the impact of pediatric tinnitus, this study proposes the development of a validated PROM set using the Patient-Reported Outcomes Measurement Information System. By identifying patient-important outcomes for different age groups and linking these outcomes to PROMIS domains, this approach ensures that the set is both comprehensive and sensitive to the diverse ways tinnitus might affect children of varying ages. While tailored to the specific domains impacted by tinnitus, the use of the generic PROMIS item banks ensures that the outcomes remain comparable across various health conditions and populations [16], and allows for an assessment of symptoms without requiring explicit attribution to a specific condition [14]. Additionally, the PROMIS item banks' availability in multiple languages and their ease of administration and interpretation make this tool widely applicable and well-suited for both clinical care and research settings. The possibility to administer the item banks using Computerized Adaptive Testing provides the additional benefit of dynamically tailoring the assessment based on the respondent's answers, thereby minimizing patient burden while increasing the accuracy of the results [16].

It is, however, important to also recognize potential challenges in using PROMIS to develop this PROM set for measuring the impact of tinnitus in children. There might be outcomes that are encountered during the creation of the patient-important outcomes list, that are not currently covered by existing PROMIS domains. In such cases, either a new PROMIS domain should be added, or additional items may need to be included within an existing item bank to comprehensively cover the domain. This process would involve collaboration with the PROMIS group, either to create a new domain with corresponding item bank or expand an existing one, both of which would require further validation for application. This process falls beyond the scope of the current study. However, the PROM set can be seen as a dynamic instrument which can be adjusted by adding or removing item banks/domains in future, also after finalization of this study and launching of the first concept of this measure. This adaptability ensures that the PROM set also remains relevant and responsive to emerging needs in health measurement.

## Acknowledgments

The authors would like to thank Caroline Terwee for her valuable advice and insights regarding the study design.

## Author contributions

**Conceptualization:** Denise Fuchten, Inge Stegeman, Adriana L. Smit.

**Methodology:** Denise Fuchten, Inge Stegeman, Adriana L. Smit.

**Supervision:** Inge Stegeman, Adriana L. Smit.

**Writing – original draft:** Denise Fuchten.

**Writing – review & editing:** Denise Fuchten, Inge Stegeman, Adriana L. Smit.

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
