## [Decision Letter · Decision Letter 0]

20 Aug 2025

PONE-D-25-31439Development of a validated PROM set for children with tinnitus using the Patient-Reported Outcomes Measurement Information System - Protocol for the 'Tinnitus in Children' (Tinc) studyPLOS ONE

Dear Dr. Fuchten,

Thank you for submitting your manuscript to PLOS ONE. After careful consideration, we feel that it has merit but does not fully meet PLOS ONE’s publication criteria as it currently stands. Therefore, we invite you to submit a revised version of the manuscript that addresses the points raised during the review process.

**ACADEMIC EDITOR: **The study is well designed. Please respond to reviewers' comments and discuss whether including clinical and radiological results would significantly alter the protocol   

We look forward to receiving your revised manuscript.

Kind regards,

Gauri Mankekar, MD,PhD,FACS

Academic Editor

PLOS ONE

Journal Requirements:

“This project is partially funded by the Dorhout Mees Stichting and through crowdfunding by Stichting Oorfonds (UMCU budgeted grant funding number 76231.92). However, the funders did not play any role in the study design and preparation of the manuscript.”

3. Thank you for stating the following in your manuscript: 

“This project is partially funded by the Dorhout Mees Stichting and through crowdfunding by Stichting Oorfonds. However, the funders did not play any role in the study design and preparation of the manuscript.”

“This project is partially funded by the Dorhout Mees Stichting and through crowdfunding by Stichting Oorfonds (UMCU budgeted grant funding number 76231.92). However, the funders did not play any role in the study design and preparation of the manuscript.”

Please include your amended statements within your cover letter; we will change the online submission 

Reviewers' comments:

Reviewer's Responses to Questions

**Comments to the Author**

1. Does the manuscript provide a valid rationale for the proposed study, with clearly identified and justified research questions?

Reviewer #1: Yes

2. Is the protocol technically sound and planned in a manner that will lead to a meaningful outcome and allow testing the stated hypotheses?

Reviewer #1: Partly

3. Is the methodology feasible and described in sufficient detail to allow the work to be replicable?

Reviewer #1: Yes

4. Have the authors described where all data underlying the findings will be made available when the study is complete?

Reviewer #1: Yes

5. Is the manuscript presented in an intelligible fashion and written in standard English?

Reviewer #1: Yes

6. Review Comments to the Author

You may also provide optional suggestions and comments to authors that they might find helpful in planning their study.

Reviewer #1: The study needs to emphasize Otological conditions such as audiological tests, examination of the ears to rule out otological disease. Radiological imaging in the form of MRI / CT Scan of the temporal bone need to be included. Exposure to noise and previous otological surgeries need to be considered.

Inclusion and exclusion criteria need to be established.

If these are not in place then the proposed study will have no impact or relevance.

7. PLOS authors have the option to publish the peer review history of their article (what does this mean? ). If published, this will include your full peer review and any attached files.

**Do you want your identity to be public for this peer review?** For information about this choice, including consent withdrawal, please see our Privacy Policy .

Reviewer #1: **Yes: ** Major rewrite of the proposal needs to be carried out.

---

## [Author Response · Author response to Decision Letter 1]

28 Sep 2025

Dear dr. Mankekar and Reviewer #1,

We thank you for your careful consideration of our manuscript and for providing constructive feedback. We have revised our manuscript to address all points raised.

Regarding the comments from the Academic Editor, we have now revised our manuscript to meet PLOS ONE’s style requirements, including file naming (comment 1). Following the editor’s guidance (comment 2), we have updated our funding statement to clarify that no additional external funding was received. The revised statement reads as follows:

“This project is partially funded by the Dorhout Mees Stichting and through crowdfunding by Stichting Oorfonds (UMCU budgeted grant funding number 76231.92). However, the funders did not play any role in the study design and preparation of the manuscript. There was no additional external funding received for this study.”

We have removed our funding statement from the manuscript and only included the updated statement in this letter, so that it can be changed in the online submission (comment 3).

Comment 4 is not applicable, as the reviewer did not suggest citing additional published work. We have verified that our reference list is complete and accurate (comment 5).

Reviewer #1 commented the following:

“The study needs to emphasize otological conditions such as audiological tests, examination of the ears to rule out otological disease. Radiological imaging in the form of MRI / CT Scan of the temporal bone needs to be included. Exposure to noise and previous otological surgeries need to be considered. Inclusion and exclusion criteria need to be established. If these are not in place then the proposed study will have no impact or relevance.”

We would like to thank the reviewer for this comment. We agree that audiological tests, otological examinations, and imaging are important when evaluating tinnitus in clinical care or studies with an etiological focus. However, the aim of our study is different. Our focus is on developing a patient-reported outcome measure (PROM) set that captures the impact of tinnitus on children’s daily lives, regardless of underlying causes or clinical characteristics.

For this reason, we apply broad inclusion criteria to ensure that the PROM set will be generalizable and relevant to the full pediatric tinnitus population. Additional clinical measures, as suggested, are therefore not required within the scope of our study. Instead, our inclusion and exclusion criteria ensure that participants are children with tinnitus who can meaningfully share their experiences (e.g., sufficient Dutch language proficiency and no diagnosed cognitive or developmental delays incompatible with age-appropriate functioning).

Once the PROM set is developed and validated, future studies could utilize this tool to explore the impact of tinnitus in children in relation to clinical characteristics. Such studies could provide valuable insights into pediatric tinnitus, but are beyond the scope of the present study.

We appreciate the opportunity to revise our manuscript and hope that the changes and clarifications address the concerns raised. Please let us know if any further modifications are required.

Sincerely,

On behalf of all authors,

Denise Fuchten

---

## [Editor Report · Decision Letter 1]

29 Sep 2025

Development of a validated PROM set for children with tinnitus using the Patient-Reported Outcomes Measurement Information System - Protocol for the 'Tinnitus in Children' (Tinc) study

PONE-D-25-31439R1

Dear Dr. Fuchten,

We’re pleased to inform you that your manuscript has been judged scientifically suitable for publication and will be formally accepted for publication once it meets all outstanding technical requirements.

Kind regards,

Gauri Mankekar, MD,PhD,FACS

Academic Editor

PLOS ONE
---

## [Editor Report · Acceptance letter]

PONE-D-25-31439R1

PLOS ONE

Dear Dr. Fuchten,

I'm pleased to inform you that your manuscript has been deemed suitable for publication in PLOS ONE. Congratulations! Your manuscript is now being handed over to our production team.

Kind regards,

on behalf of

Dr. Gauri Mankekar

Academic Editor

PLOS ONE